# Efficacy of a Deep Learning Convolutional Neural Network System for Melanoma Diagnosis in a Hospital Population

**DOI:** 10.3390/ijerph19073892

**Published:** 2022-03-24

**Authors:** Manuel Martin-Gonzalez, Carlos Azcarraga, Alba Martin-Gil, Carlos Carpena-Torres, Pedro Jaen

**Affiliations:** 1Service of Dermatology, Hospital Universitario Ramón y Cajal, 28034 Madrid, Spain; carlos.azcarraga95@gmail.com (C.A.); pedro@pjaen.com (P.J.); 2Instituto Ramón y Cajal de Investigación Sanitaria, 28034 Madrid, Spain; 3Ocupharm Research Group, Department of Optometry and Vision, Faculty of Optics and Optometry, Complutense University of Madrid, 28037 Madrid, Spain; amarting@ucm.es (A.M.-G.); ccarpena@ucm.es (C.C.-T.)

**Keywords:** melanoma, skin cancer, oncology, artificial intelligence, deep learning

## Abstract

(1) Background: The purpose of this study was to evaluate the efficacy in terms of sensitivity, specificity, and accuracy of the quantusSKIN system, a new clinical tool based on deep learning, to distinguish between benign skin lesions and melanoma in a hospital population. (2) Methods: A retrospective study was performed using 232 dermoscopic images from the clinical database of the Ramón y Cajal University Hospital (Madrid, Spain). The skin lesions images, previously diagnosed as nevus (*n* = 177) or melanoma (*n* = 55), were analyzed by the quantusSKIN system, which offers a probabilistic percentage (diagnostic threshold) for melanoma diagnosis. The optimum diagnostic threshold, sensitivity, specificity, and accuracy of the quantusSKIN system to diagnose melanoma were quantified. (3) Results: The mean diagnostic threshold was statistically lower (*p* < 0.001) in the nevus group (27.12 ± 35.44%) compared with the melanoma group (72.50 ± 34.03%). The area under the ROC curve was 0.813. For a diagnostic threshold of 67.33%, a sensitivity of 0.691, a specificity of 0.802, and an accuracy of 0.776 were obtained. (4) Conclusions: The quantusSKIN system is proposed as a useful screening tool for melanoma detection to be incorporated in primary health care systems.

## 1. Introduction

Cutaneous melanoma is the most common melanoma subtype [1] and one of the most aggressive forms of skin cancer, causing about 90% of deaths associated with cutaneous tumors [2] and about 1–2% of all cancer-related deaths [3,4]. The cutaneous melanoma prognostic is influenced by several factors such as early diagnosis. In this sense, melanomas detected in the advanced stage have a worse prognosis, being below 15% of the five-year survival ratio in contrast to melanomas detected in the initial stages which elevate survival rate to 95% [5].

Skin biopsy remains the most important step to confirm a definitive diagnosis of cutaneous melanoma. Currently, cutaneous melanoma is initially diagnosed via visual skin inspection by the ABCDE assessment, which evaluates characteristics of the skin lesions such as asymmetry (A), borders (B), color (C), diameter (D), and evolving (E) [6]. The ABCDE assessment is followed by a dermoscopy analysis which improves diagnosis accuracy from 60% (by clinician’s visual inspection) to 75–84% [5,7], reaching a sensitivity of 89% and a specificity of 79% [8].

Moreover, in recent years there has been a rise in the use of several techniques of automated computer image analysis to provide an improvement in diagnostic accuracy and reproducibility for melanoma screening regarding the clinical diagnosis results from dermoscopic images [5,9,10]. In this sense, artificial intelligence based on deep learning models, specifically convolutional neural networks (CNNs), has been used in several studies to test its ability in melanoma diagnosis from dermoscopic images, showing results in effectivity comparable to dermatologists’ prediction, but in a shorter amount of time, thus allowing reduction of costs [5,9,10]. However, the main limitation in melanoma diagnosis is the need to train the CNN with a large amount of data.

As reported in a previous study, our research group trained a deep learning CNN with 37,688 biopsy-verified images from the International Skin Imaging Collaboration (ISIC) Archive to classify images into either benign or malignant skin lesions, to reach acceptable sensitivity and specificity values [11]. Nevertheless, no previous study focused on the use of a deep learning-based method as screening tools in primary health care rather than diagnosis techniques in specialized dermatology consultation. In addition, most of these studies trained and tested their CNN tool with images from public databases.

The purpose of this study was to evaluate the efficacy in terms of sensitivity, specificity, and accuracy of the quantusSKIN system, a new clinical tool based on deep learning, to distinguish between benign skin lesions and melanoma in a Spanish hospital population as an innovative screening tool. At the same time, the secondary purpose was to evaluate the ability of retraining of the quantusSKIN system with a specific hospital population to improve its detection. The motivation of this study was to generate scientific evidence on the efficacy of the quantusSKIN system to be successfully incorporated in clinical practice.

## 2. Materials and Methods

### 2.1. Study Design and Sample

A retrospective study was carried out in the Hospital Universitario Ramón y Cajal (Madrid, Spain). The ethics committee of the same hospital approved the protocol of the study (code 159/21). The clinical database of dermoscopic images from the Service of Dermatology of the hospital was used. The images were captured by using a digital camera coupled to three different manual dermatoscopes DermLite II PROHR 3GEN, DL200 HR, and DL4W (3Gen Inc.; San Juan Capistrano, CA, USA). Additionally, a portion of images lower than 5% was captured with the digital dermoscopy system FotoFinder (FotoFinder; Bad Birnbach, Germany).

A total of 571 dermoscopic images of 513 different white skin patients were obtained, from which 339 were training images and 232 were testing images. These images were previously diagnosed as nevus or melanoma by different dermatologists of the hospital. The melanoma diagnosis was confirmed from the histopathologic examination after removing the skin lesion in all cases, as the gold standard. Different types of melanomas were included, both in situ and invasive, except those amelanotic, subungual, mucosal, or ocular melanomas. The melanocytic nevi with clinical features of atypical nevi were diagnosed by dermoscopy according to the absence of changes for a minimum follow-up period of 1 year, except in 50 skin lesions where a histopathologic analysis was necessary.

### 2.2. QuantusSKIN System

The quantusSKIN system (Transmural Biotech; Barcelona, Spain) is a web-based deep learning tool for melanoma diagnosis based on automated analysis of dermoscopic images. The images incorporated into the system were well-centered and isolated skin lesions occupying at least two-thirds of image dimensions, and with no light filters, measurement marks over the skin lesion, or hair covering this lesion (Figure 1).

Originally, the quantusSKIN system was trained with 37,688 training images obtained from the public database ISIC Archive 2019 and 2020 [11]. In this current study, this network was retrained with the additional 339 training images (196 nevi and 143 melanomas) of 291 different patients obtained from the clinical practice of the Hospital Universitario Ramón y Cajal. The mean age of these patients was 50.99 ± 20.99 years (range 3 to 91 years).

For the image segmentation, a fully convolutional network (FCN-8s) [12] was used to automatically delimitate the skin lesion in each image by using the artificial intelligence platform PROTO (Transmural Biotech). Furthermore, with the same platform, a convolutional network with a variant of the inception algorithm was adapted for the image classification [13]. The skin lesion classifier was trained using SoftMax classification with cross-entropy loss and Adam optimization. For network training, the maximum number of epochs was 30, early stopping was established in case the loss was not improved for 10 consecutive epochs, and a cosine decay starting at 10^−4^, a batch size of 64, and a weight decay of 0.9 were given. At each batch, the training images were randomly flipped, cropped between 0–10%, translated from 0–10 pixels, and rotated between −90, 90 degrees.

Therefore, because the quantusSKIN system is a tool for binary diagnosis (benignant/malignant skin lesion), its optimum diagnostic threshold (expressed in probabilistic percentage) was established based on the analysis of the 232 testing images.

### 2.3. Statistical Analysis

The statistical analysis was carried out with the software SPSS Statistics 24 (IBM; Chicago, IL, USA). The normality of the distributions was assessed using the Shapiro–Wilk test. Once the normality of the variables was confirmed, the Student’s *t*-test for independent samples was selected to compare the diagnostic threshold offered by the quantusSKIN between the nevus and melanoma groups, in addition to the age. Additionally, the chi-square test was performed to analyze the association between gender and type of skin lesion in the total sample. A statistical significance of 95% was established (*p* < 0.05), while the results are shown as mean ± standard deviation.

A receiver operating characteristics (ROC) analysis was carried out to quantify the sensitivity and specificity of the quantusSKIN to diagnose melanoma in comparison with the histological diagnosis as the gold standard. The area under the curve (AUC), sensitivity, specificity, accuracy, positive predictive value (PPV), negative predictive value (NPV), and F1 score were reported for two different error metrics: maximum F1 score and specificity higher than 0.80 together with maximum sensitivity. The variables were calculated as follows:Sensitivity = true positive/(true positive + false negative),(1)
Specificity = true negative/(true negative + false positive),(2)
Accuracy = (true positive + true negative)/total sample,(3)
PPV = true positive/(true positive + false positive),(4)
NPV = true negative/(true negative + false negative),(5)
F1 score = 2 × ((PPV × sensitivity)/(PPV + sensitivity)), which represents the harmonic mean of PPV and sensitivity.(6)

## 3. Results

A total of 232 skin lesions of 222 different patients (45.56 ± 19.78 years, range 2 to 95 years) were diagnosed as nevus or melanoma according to the established clinical criteria. Table 1 summarizes the characteristics of either group. The mean age of the nevus group was 40.91 ± 17.83 years, while the melanoma group was 60.53 ± 18.39 years, statistically significant differences being seen between both groups (*p* < 0.001). Conversely, the chi-square test found no association between gender and type of skin lesion (*p* = 0.060).

### 3.1. Efficacy after Re-Training

Concerning the results of the testing images analysis, the mean diagnostic threshold was statistically lower (*p* < 0.001) in the nevus group (27.12 ± 35.44%) compared with the melanoma group (72.50 ± 34.03%). On the other hand, Figure 2 represents the ROC curve, which obtained an AUC value of 0.813, while the results of the optimum threshold of the quantusSKIN for diagnosing melanoma derived from the ROC analysis are summarized in Table 2. To obtain a maximum F1 score (0.614), the quantusSKIN increased its optimum diagnostic threshold to 53.51%. Finally, to reach a specificity higher than 0.800, together with the maximum sensitivity possible (0.691), the optimum diagnostic threshold of the quantusSKIN for melanoma diagnosis was 67.33%.

From a clinical viewpoint, establishing a diagnostic threshold of 67.33% after retraining represents that 30.9% of melanomas are not diagnosed by the quantusSKIN system (false negatives) and 19.8% of nevi are considered as melanomas (false positives).

### 3.2. Efficacy before Re-Training

Figure 2 also shows the ROC curve of the 232 skin lesions’ analysis before retraining the quantusSKIN system with the 339 training images. The AUC of this ROC curve (0.799) was slightly lower than the one with retraining. For a diagnostic threshold of 65.57%, which was the optimum value for reaching a specificity higher than 0.800 and maximum sensitivity, the specificity was the same as before (0.802). However, the sensitivity and accuracy decreased to 0.655 and 0.767, respectively. The values of PPV (0.507), NPV (0.882), and F1 score (0.571) were also lower compared with the results after retraining.

From a clinical viewpoint, establishing a diagnostic threshold of 65.57% before re-training represents that 35.5% of melanomas are not diagnosed by the quantusSKIN system (false negatives) and 19.8% of nevi are considered as melanomas (false positives).

## 4. Discussion

This study analyses the efficacy of a new version of the quantusSKIN system to distinguish between benign skin lesions and melanomas in terms of sensitivity, specificity, and accuracy. This clinical screening tool, which had been previously trained with 37,688 images, was subsequently retrained during this assay with 339 new training images of different patients from a Spanish Hospital. The clinical aim of the quantusSKIN system is to be incorporated into primary health care to minimize the number of false positives referred to a physician specialist. In this sense, the choice of its optimum diagnostic threshold focused on maximizing system sensitivity (higher than 0.800), the reason being that the optimum threshold of the quantusSKIN system for melanoma diagnosis was established at 67.33%, slightly lower than the mean value obtained by the melanoma group (72.50%). At the same time, this diagnostic threshold would also allow obtaining accuracy and NPV close to 0.800 and 0.900, respectively. The purpose of finding a high NPV is to enable the quantusSKIN use in primary health care during a preliminary screening, referring only patients with a suspected diagnosis of melanoma to specialized dermatologists. These facts could help reduce existing waiting lists and associated medical costs.

The accuracy and specificity reported in the current study are in agreement with Kaur et al., who previously reported accuracy of 0.830 and specificity of 0.839 with their CNN model, but with better sensitivity results than ours [14]. The purpose of their study was to find high sensitivity values since they were looking for a diagnosis tool, while the quantusSKIN system is intended to become a screening tool, so it was necessary to find as high of a specificity as possible to reduce false positives and to increase NPV. Similarly, Brinker et al. reported a sensitivity of 0.682 with their trained CNN, which was better than junior and board-certified dermatologist results (with sensitivity and specificity under 0.700 and 0.660, respectively), which was in agreement with the outcomes of our CNN trained model [10]. At the same time, Haenssle et al. compared sensitivity and specificity between their CNN model and a large number of dermatologists’ diagnostic performances, finding better dermatologist outcomes than Brinker et al. In their study, Haenssle et al. reported 0.866 of sensitivity and 0.757 of specificity of the dermatologists versus 0.950 of sensitivity and 0.825 of specificity of the CNN algorithm training [9], which were better results than the current study. Other authors have also reported better sensitivity values than ours (higher than 0.900) but at the expense of their specificity results [15].

On the other hand, in the current study, it would be possible to decrease the diagnostic threshold at 53.51% (maximum F1 score) to increase sensitivity by 9% (from 0.692 to 0.782). However, this threshold would decrease specificity by 4% (from 0.802 to 0.763), which is not clinically tolerated from a primary health care perspective since the same percentage of false positives would be wrongly referred to dermatology consultation, accompanied by increased costs and waiting lists.

Concerning the question of whether deep learning CNN models should be retrained with images of the same dermatological consultation to improve melanoma diagnosis, the quantusSKIN system improved its efficacy after being retrained with the 339 training images. After the retraining, the quantusSKIN obtained better values of sensitivity (from 0.655 to 0.691), accuracy (from 0.767 to 0.776), and the rest of the parameters, except specificity. This fact would show that CNN models should not only be trained with images from public databases [11], but also from the medical consultations where they are used, considering, for example, the different skin phenotypes in each medical population. In fact, the influence of skin color on the efficacy of the quantusSKIN system was not investigated in the current study, which supposed one of its main limitations.

Moreover, this study presented other limitations that should be improved in future studies. In this sense, prospects may involve the retraining of the system with real images taken with conventional devices, such as adapted smartphones. In such cases, it would be interesting to check its reproducibility between images taken by different devices and to compare its outcomes in terms of specificity, sensitivity, and accuracy with our previous results obtained from dermoscopy images. Additionally, differences in image quality should be checked in the future, considering changes in lighting conditions, filters, and magnification, in addition to different characteristics of the skin lesions such as skin color, presence or absence of hair. At the same time, new training with images of different types of non-cancerous pigmented skin lesions such as seborrheic keratosis, for example, would be necessary to improve the useful screening tool to distinguish with greater accuracy between a higher number of benign and malign lesions. Furthermore, it would be interesting to compare the new CNN model results with dermatologists’ diagnostic performance and with physicians’ diagnostic performance from primary health care to demonstrate its usefulness as a screening tool.

## 5. Conclusions

In a real Spanish hospital population, the quantusSKIN system showed a sensitivity of 0.691, specificity of 0.802, and accuracy of 0.776 to diagnose melanoma for an optimum diagnostic threshold of 67.33%. Therefore, it seems appropriate that the quantusSKIN system should be incorporated as a screening tool in primary health care systems. Additionally, it has been proposed that retraining CNN models with specific hospital images could improve the efficacy of these systems to diagnose melanoma compared with the use of training images from public databases only.

## Figures and Tables

**Figure 1 ijerph-19-03892-f001:**
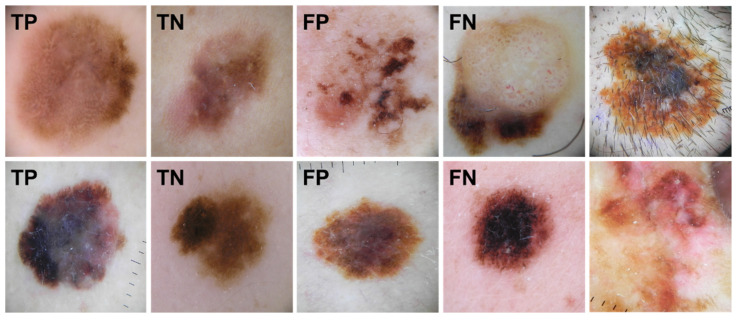
Representative images of different skin lesions diagnosed by the quantusSKIN system: true positive (TP), true negative (TN), false negative (FN), and false positive (FP), in addition to two images not recommended to analyze (right column): hair covering the skin lesion (upper) and skin lesion occupying the full image dimensions (lower).

**Figure 2 ijerph-19-03892-f002:**
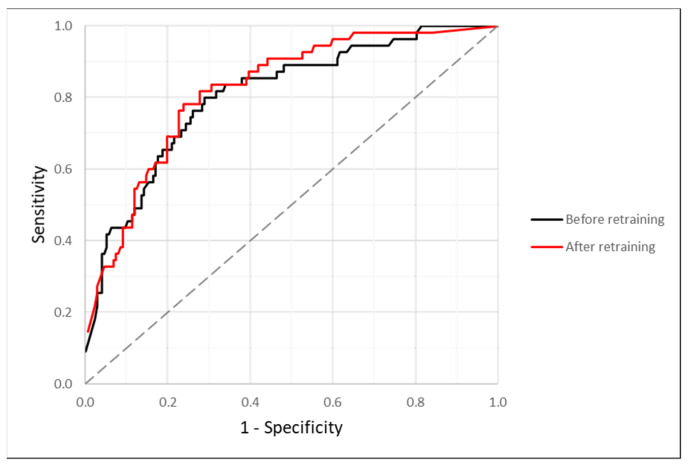
Receiver operating characteristics (ROC) curves obtained from the analysis of the 232 testing images by using the quantusSKIN system before the retraining with the additional 339 training images (black) and after this retraining (red).

**Table 1 ijerph-19-03892-t001:** Demographic characteristics of the participants in the study and their skin lesion locations.

Parameter	Nevus Group	Melanoma Group
Sample (*n*)	177	55
Age (years)	40.91 ± 17.83	60.53 ± 18.39
Gender (F/M)	121/56	30/25
Skin lesion location (*n*, %)		
Scalp	0 (0.0%)	1 (1.8%)
Face	5 (2.8%)	6 (10.9%)
Neck	2 (1.1%)	2 (3.6%)
Trunk	144 (81.4%)	26 (47.3%)
Upper extremity	10 (5.7%)	6 (10.9%)
Lower extremity	10 (5.7%)	11 (20.0%)
Hand	1 (0.6%)	0 (0.0%)
Foot	4 (2.3%)	1 (1.8%)
Vulvar skin	1 (0.6%)	1 (1.8%)
Foreskin	0 (0.0%)	1 (1.8%)

**Table 2 ijerph-19-03892-t002:** Efficacy of the quantusSKIN system for melanoma diagnosis after its retraining in terms of sensitivity, specificity, accuracy, PPV, NPV, and F1 score for the optimum diagnostic threshold established with two different error metrics. Additionally, the efficacy data of other existing deep learning algorithms recently reported in the scientific literature is summarized.

Error Metric/Study	Diagnostic Threshold (%)	Sensitivity	Specificity	Accuracy	PPV	NPV	F1 Score	F2 Score
Maximum F1 score	53.51	0.782	0.763	0.767	0.506	0.918	0.614	0.705
Specificity > 0.800 andmaximum sensitivity	67.33	0.691	0.802	0.776	0.521	0.893	0.594	0.648
Haenssle et al. [9]	-	0.950	0.825	-	-	-	-	-
Brinker et al. [10]	-	0.682	-	-	-	-	-	-
Kaur et al. [14]	-	0.830	0.839	0.830	-	-	-	-

## Data Availability

The data used to support the findings of this study are available from the corresponding author upon request.

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
