# Peer review of "Efficacy of a Deep Learning Convolutional Neural Network System for Melanoma Diagnosis in a Hospital Population"

_ijerph, 2022, doi:10.3390/ijerph19073892_

Round 1
Reviewer 1 Report
- in the legend of the figure 1 (Line 89): false positive (FN),: FN must be FP
- in the introduction, the authors state the following: "cutaneous melanoma is initially diagnosed via visual skin inspection...". It would be interesting to add more details about this visual inspection by highlighting the ABCDE melanoma algorithm...
Author Response
Response to Reviewer #1 in this letter and changes into the manuscript are done in Red and to Reviewer #2 in Blue, Reviewer #3 in Green, and Reviewer #4 in Purple.
Thank you for your time reviewing the manuscript and your favorable evaluation and comments to improve it. We have considered your comments.
· In the legend of the figure 1 (Line 89): false positive (FN): FN must be FP
Thank you for your comment. We have corrected this mistake in Figure 1.
· In the introduction, the authors state the following: "cutaneous melanoma is initially diagnosed via visual skin inspection...". It would be interesting to add more details about this visual inspection by highlighting the ABCDE melanoma algorithm...
Thank you for your comment. We have added this information in the Introduction:
“Currently, cutaneous melanoma is initially diagnosed via visual skin inspection by the ABCDE assessment, which evaluates characteristics of the skin lesions such as asymmetry (A), borders (B), color (C), diameter (D), and evolving (E) [6]. The ABCDE assessment is followed by a dermoscopy analysis…
[6] Tsao, H.; Olazagasti, J.M.; Cordoro, K.M.; Brewer, J.D.; Taylor, S.C.; Bordeaux, J.S.; Chren, M.M.; Sober, A.J.; Tegeler, C.; Bhushan, R.; et al. Early detection of melanoma: Reviewing the ABCDEs. Journal of the American Academy of Dermatology 2015, 72, 717-723, doi:10.1016/j.jaad.2015.01.025.”

Reviewer 2 Report
The novelty of this paper is limited. It only uses new data to verify the effectiveness of the quantusSKIN system, while it does not propose new methods and does not contribute to the field. At the same time, the motivation of this paper is not clear. It is well known that deep learning methods have better performance than machine learning-based methods. Why do we evaluate the effectiveness of CNN-based methods?
Author Response
Response to Reviewer #1 in this letter and changes into the manuscript are done in Red and to Reviewer #2 in Blue, Reviewer #3 in Green, and Reviewer #4 in Purple.
· The novelty of this paper is limited. It only uses new data to verify the effectiveness of the quantusSKIN system, while it does not propose new methods and does not contribute to the field. At the same time, the motivation of this paper is not clear. It is well known that deep learning methods have better performance than machine learning-based methods. Why do we evaluate the effectiveness of CNN-based methods?
Thank you for your revision and comments. We agree with you on the idea that the current study is not the greater novelty in the field since there are other artificial intelligence-based methods for melanoma diagnosis. However, like the rest of commercially available systems, the quantusSKIN requires scientific evidence to be successfully validated on clinical images and used for clinicians in their daily practice, the reason why this study was performed.
“The motivation of this study was to generate scientific evidence on the efficacy of the quantusSKIN system to be successfully incorporated in clinical practice.” has been incorporated in the Introduction.
On the other hand, we think that Reviewer #2 misunderstood the principle of the quantusSKIN system based on a mistake that we made in the first sentence of the quantusSKIN system section. We incorrectly redacted machine learning instead of deep learning, so this mistake has been corrected.
The quantusSKIN system is a deep learning system where a CNN was trained with dermoscopic images, so we did not use machine learning procedures (the parts where we mention this information have been blue highlighted in the manuscript). The only input that was introduced in the CNN was the previous diagnosis of nevus or melanoma, but no information about the skin lesions characteristics was used.

Reviewer 3 Report
This paper describes the efficacy of the quantusSKIN system in terms of sensitivity, specificity, and accuracy.
The quantusSKIN system is a new clinical tool as an innovative screening tool based on deep learning and it is used to distinguish between benign skin lesions and melanoma.
Some suggestions to improve the paper are:
1) It seems that better experimental results can be derived by adding the part about race(whether black, yellow, or white) among the parameters in Table1.
2) It would be better if Table2 shows the comparative performance with other existing algorithms.
3) For skin color, it seems necessary to build learning data according to various lighting conditions.
Author Response
Response to Reviewer #1 in this letter and changes into the manuscript are done in Red and to Reviewer #2 in Blue, Reviewer #3 in Green, and Reviewer #4 in Purple.
· This paper describes the efficacy of the quantusSKIN system in terms of sensitivity, specificity, and accuracy. The quantusSKIN system is a new clinical tool as an innovative screening tool based on deep learning and it is used to distinguish between benign skin lesions and melanoma.
We would like to thank your suggestions to improve the manuscript.
· Some suggestions to improve the paper are:
· 1) It seems that better experimental results can be derived by adding the part about race (whether black, yellow, or white) among the parameters in Table 1.
· 3) For skin color, it seems necessary to build learning data according to various lighting conditions.
Thank you for your comments. We agree that race and skin color are two very interesting points that, unfortunately, we could not consider due to the low prevalence of non-Caucasian or non-white skin patients in our hospital database.
More information about these methodological aspects and study limitations have been added to the manuscript:
Since the concept of “race” could be controversial, we have added this information to Study Design and Sample section: “A total of 571 dermoscopic images of 513 different white skin patients were obtained”.
“…the different skin phenotypes in each medical population. In fact, the influence of skin color on the efficacy of the quantusSKIN system was not investigated in the current study, which supposes one of its main limitations.
Moreover, this study presented other limitations that should be improved in future studies…” has been modified in the Discussion.
· 2) It would be better if Table 2 shows the comparative performance with other existing algorithms.
Thank you for your suggestion. We have added data about other algorithms evaluated in the scientific literature in Table 2.

Reviewer 4 Report
The authors presents a method to distinguish between skin lesions and melanoma in a hospital population using deep neural network system. The results shos an accuracy of 77.6 %, and in my opinion is a good paper.
Author Response
Response to Reviewer #1 in this letter and changes into the manuscript are done in Red and to Reviewer #2 in Blue, Reviewer #3 in Green, and Reviewer #4 in Purple.
· The authors presents a method to distinguish between skin lesions and melanoma in a hospital population using deep neural network system. The results show an accuracy of 77.6 %, and in my opinion is a good paper.
We would like to express our gratitude to this reviewer for his/her favorable opinion.

Round 2
Reviewer 2 Report
My concerns have been well solved. I agree to accept this paper.